# Rational Design and Porosity of Porous Alumina Ceramic Membrane for Air Bearing

**DOI:** 10.3390/membranes11110872

**Published:** 2021-11-12

**Authors:** Jianzhou Du, Duomei Ai, Xin Xiao, Jiming Song, Yunping Li, Yuansheng Chen, Luming Wang, Kongjun Zhu

**Affiliations:** 1School of Materials Science and Engineering, School of Mechanical Engineering, Yancheng Institute of Technology, Yancheng 224051, China; dujianzhou123@163.com (J.D.); 15052323525@163.com (D.A.); xiaoxin0627@163.com (X.X.); jm68742@163.com (J.S.); liyp@ycit.edu.cn (Y.L.); chenys@ycit.edu.cn (Y.C.); 2State Key Laboratory of Mechanics and Control of Mechanical Structures, Nanjing University of Aeronautics and Astronautics, Nanjing 210016, China

**Keywords:** porous alumina ceramics, inorganic membranes, porosity, modeling, air bearing

## Abstract

Air bearing has been widely applied in ultra-precision machine tools, aerospace and other fields. The restrictor of the porous material is the key component in air bearings, but its performance is limited by the machining accuracy. A combination of optimization design and material modification of the porous alumina ceramic membrane is proposed to improve performance within an air bearing. Porous alumina ceramics were prepared by adding a pore-forming agent and performing solid-phase sintering at 1600 °C for 3 h, using 95-Al_2_O_3_ as raw material and polystyrene microspheres with different particle sizes as the pore-forming agent. With 20 wt.% of PS50, the optimum porous alumina ceramic membranes achieved a density of 3.2 g/cm^3^, a porosity of 11.8% and a bending strength of 150.4 MPa. Then, the sintered samples were processed into restrictors with a diameter of 40 mm and a thickness of 5 mm. After the restrictors were bonded to aluminum shells for the air bearing, both experimental and simulation work was carried out to verify the designed air bearing. Simulation results showed that the load capacity increased from 94 N to 523 N when the porosity increased from 5% to 25% at a fixed gas supply pressure of 0.5 MPa and a fixed gas film thickness of 25 μm. When the gas film thickness and porosity were fixed at 100 μm and 11.8%, respectively, the load capacity increased from 8.6 N to 40.8 N with the gas supply pressure having been increased from 0.1 MPa to 0.5 MPa. Both experimental and simulation results successfully demonstrated the stability and effectiveness of the proposed method. The porosity is an important factor for improving the performance of an air bearing, and it can be optimized to enhance the bearing’s stability and load capacity.

## 1. Introduction

Due to its simple structure, high temperature resistance, high precision, high energy efficiency and zero pollution, the air bearing has been widely applied in ultra-precision engineering, space technology, micro engineering and so on [1,2,3]. The restrictor that controls the flow velocity is the key component in the air bearing, but a classic orifice-type restrictor limits its stability and precision due to a gas hammer effect [4]. Porous restriction can achieve a more uniform pressure distribution on the bearing’s surface, and as such is receiving more and more attention [5,6,7,8,9]. 

The performance of the air bearing depends on the properties of porous materials, such as porous metals [10], porous graphite [11] and porous ceramics [12]. Porous metals are manufactured by drilling many small holes on metal, and their performance is largely limited by machining precision. The debris from machine work may block the drilled holes and worsen the performance of the air bearing. Porous graphite has the advantage of machining performance and lubrication, but its high cost limits its practical application. Due to their stability, high porosity, wear resistance, long service life and high rigidity, porous ceramics membranes are more attractive options for air bearing improvement [13,14,15].

Structural optimization and material design are the two typical strategies that have been developed to improve the properties of the restrictor. For example, a downsized aerostatic circular thrust bearing with a single feed hole was analyzed using computational fluid dynamics (CFD) [16]. In addition, a numerical calculation method, via a finite element method and a proportional division method, was proposed to optimize the angular stiffness of aerostatic bearings [17]. In another related work, the nonlinear behaviors of the aerostatic bearing–rotor system were theoretically and experimentally investigated [18]. In Ref. [19], a mathematical model of a circular flat pad aerostatic bearing with a single central orifice-type restrictor was developed for design optimization.

The improvement of porous material properties is an effective way to upgrade the performance of air bearings. Given that porous metals are limited by machine work and porous graphite is too expensive, porous ceramics are more attractive for the future design of air bearings. In Ref. [20], the porous Al_2_O_3_ ceramic membrane, with pores introduced by the polymethyl methacrylate, showed bimodal pore size distribution. Rezaee et al. [21] prepared porous alumina ceramics using corn starch as the pore-forming agent via a sol–gel method, and the results showed that pores have near spherical morphology with uniform size and distribution and good connectivity. The 40 wt.% starch sample doped with niobium retained a maximum porosity of 57% and a compressive strength of 60 MPa. Ribeiro et al. [22] used the rice husk as a pore-forming agent to prepare porous alumina ceramic membranes with a total porosity ranging from 24% to 54%. In Ref. [23], the effect of core–shell microspheres as a pore-forming agent on the properties of porous alumina ceramics was discussed. With an increased number of microspheres, the closed porosity of alumina ceramic membranes ranges from 5.82% to 14.5%, while their average flexural strength changes from 291.98 MPa to 97.05 MPa.

Note that these previous attempts to improve the air bearing were focused on either optimization design or material modification, while the combination of these two strategies has rarely been involved. In this work, optimization design was firstly carried out by finite element analysis to obtain the optimum porosity, and then the restrictor material was sintered with a pore-forming agent to meet the calculated value of porosity. The proposed strategy that combined both optimization design and material modification was further verified by simulation and experimental work, which successfully demonstrated the effectiveness and stability of the designed air bearing. 

## 2. Experimental Procedure

### 2.1. Model Configuration and Grid Meshing

The main structure of the air bearing consists of the bearing body and the restrictor, and the latter is traditionally made of porous metal. A novel design with a porous alumina ceramic membrane is proposed for the restrictor in the air bearing. When porous ceramic membranes are used, the gas flow is more uniform and the proposed air bearing is more stable. As shown in Figure 1, the compressed air enters the inlet of the air bearing and then flows out through the restrictor. When the compressed air comes out of the porous alumina ceramic, the air flow pushes the guiding surface, and its reaction force floats the air bearing. The air pressure decreases around the restrictor, and a gas film is generated between the porous alumina ceramic and the guiding surface. The thickness of the gas film can be tuned by the central air pressure, which further changes with the load of the proposed air bearing. That is, the air bearing can float with a certain load and avoid contact between the bearing body and the guiding surface [24,25,26]. Due to the lubrication provided by the flowing air, the air bearing faces less friction and abrasion and can prolong the service life of the system. 

The geometric model of the proposed air bearing is illustrated in Figure 2a, where the blue part represents the bearing body, and the gray part indicates the porous alumina ceramic restrictor. Using the design parameters in Table 1, a fluid model is developed in ANASYS/Fluent software. As shown in Figure 2b, the fluid model contains three parts, including an air chamber restrictor of porous alumina ceramic and a gas film. Because the thickness of the gas film is the key parameter in this simulation, the thickness of the meshing is set to two microns, as shown in Figure 2c.

To simplify the calculation in ANASYS/Fluent, it was assumed that:The flowing air in this simulation is a kind of ideal gas.The flowing gas between the air bearing and the guiding surface is laminar.All walls are smooth, and their damping effect on air is ignored.The porous alumina ceramic is isotropic, with the same permeability coefficient in all directions.

### 2.2. Fabrication of Porous Alumina Ceramics Membrane

The alumina powders doped with 1.5 wt.% Yb_2_O_3_, 1.5 wt.% MgO and 2 wt.% SiO_2_ (named 95-Al_2_O_3_), were selected as the main raw materials to prepare the porous alumina ceramic membrane. Polystyrene microspheres (PS, Dongguan Baolimei Plastic Co., Ltd., Dongguan, China) with different specifications (PS10, PS30, PS50, PS100) were adopted as the pore-forming agent. The particles of PS samples were measured individually via the laser granularity analyzer (Type. LS13320, Beckman Coulter, Inc., CA, USA) with the dry powder system, and the precise particle size distributions were developed by the Particle Size Analyzer V6.01. The median particle sizes of PS10, PS30, PS50 and PS100 were 8.7 μm, 20.2 μm, 39.8 μm and 86.9 μm, respectively, as shown in Figure 3. The 95-Al_2_O_3_ raw powder was mixed with 20 wt.% of PS and 5 wt.% of polyvinyl alcohol solution (PVA217, Kuraray Co., Ltd., Osaka, Japan) with a viscosity of 30 mPa·s, the result of which was then added as the binder. After ball milling treatment, the slurry was dried, ground and then pressed in a designed mold. The green bodies were heated up to 1600 °C holding for 3 h with a ramping rate of 2 °C/min to achieve a porous alumina ceramic membrane.

### 2.3. Characterization

The density (*ρ*) and porosity (*P*) of the sintered samples were measured by way of the Archimedes method with deionized water as the immersion medium, using an automatic electronic densitometer (Type ZMD-2, Shanghai Fangrui Instrument Co. Ltd., Shanghai, China). A scanning electron microscope (SEM, Nova Nano SEM 450, FEI Co., OR, USA) was employed to observe the microstructures of the samples, and the pore size distribution was analyzed by way of Nano Measure software (Ver. 1.2, Fudan University, China). The bending strength (*σ_f_*) of samples with dimensions of 50 mm × 5 mm × 5 mm was tested based on the three-point bending method using a universal testing machine (Type. RG-4010, Shenzhen Reger Co. Ltd., Shenzhen, China) with a span length of 24 mm and a loading rate of 5 mm/min. The arithmetic mean of the five groups of results was finally presented. The calculation formula was as follows:$$\sigma_{f}=\frac{3FL}{2bh^{2}}$$
where *σ_f_* is the bending strength (MPa), *F* is the load at fracture, *L* is the length of span (mm), *b* is the specimen cross-section width (mm) and *h* is the specimen cross section height (mm). 

## 3. Results and Discussion

### 3.1. Simulation Analysis of Porosity

The simulated results of gas pressure distribution are shown in Figure 4, where the thickness of the gas film (*h*), air supply pressure (*P*s) and environmental temperature (T) were set as 25 μm, 0.5 MPa and 300 K, respectively. It can be found that the gas pressure on the bearing surface is symmetrically distributed and decreases gradually from the center to the surrounding. At a porosity of 5%, the pressure on the bearing surface is relatively small. The stable area of gas pressure is less than 50% at a porosity of 10%. The stable area covers over 90% when the porosity is 25%. 

As shown in Figure 4f, when the porosity increases from 5% to 10%, 15%, 20% and 25%, the bearing capacity increases from 94 N to 338 N, 451 N, 503 N and 523 N, respectively. It should be noted that the stable area of gas pressure is too small to support the load when the porosity is in the range of 5%~10%. Therefore, the porosity of the porous alumina ceramic should be larger than 10%. In short, the relationship between porosity and bearing capacity of the air bearing is approximately linear. The bearing capacity increases with the increase of porosity.

### 3.2. Microstructure and Performance

The fracture micromorphology of the sintered samples is shown in Figure 5. With the addition of PS of different dimensions, pores of different sizes were formed on the surface of the porous alumina ceramic membrane. The pore formation during the firing process of the porous ceramic membrane is dependent on the shape of the PS. Both the amount and diameter of these pores increases with those of the PS. As shown in Figure 5a, many pores are formed when PS10 are added, but their size is much smaller, 2.7 μm in diameter. The pores are uniformly distributed on the fracture surface, and there is no obvious pore overlap. The porous Al_2_O_3_ ceramic samples sintered with larger PS (PS30) are shown in Figure 5b. Their diameter is relatively small, and it varies in a wide range. After analysis by way of Nano Measurer software, an average diameter of 6.2 μm was obtained, with 50% of them in the range of 5 μm~7 μm. 

The pore diameters in Figure 5c are in the range of 10 μm~30 μm, and their average value is 17.8 μm. The obvious overlap of these pores is related to the larger size of the PS (PS50), which easily leads to an uneven distribution of pore-forming agents. As shown in Figure 5d, the shape of the pores is relatively regular when formed using PS100 as pore-forming agents, with which the average diameter of pores is determined to be 49.9 μm. In the case of the same mass, with the increase in PS size, the number of PS is relatively reduced while the porosity is increased instead. This is because the spaces generated by particle accumulation will form micropores after sintering and the existence of pores increases the porosity. The articles also illustrate this point [27,28]. When porous ceramic membranes are sintered at high temperature, the green body shrinks significantly, resulting in the decrease in or even disappearance of pores. In general, the shrinkage of ceramic influences the pore size of a porous ceramic membrane but has a smaller effect on the uniformity of pore size distribution [29], and the pore size distribution is basically consistent with that of the pore-forming agent. 

Figure 6 illustrates the density, porosity and bending strength of a sintered porous Al_2_O_3_ ceramics membrane prepared with PS of different diameters. It can be found that the sample’s density slowly decreased from 3.41 g/cm^3^ to 3.20 g/cm^3^ and porosity increased from 6.2% to 11.8% when the size of the pore-forming agent gradually increased. The bending strength of porous alumina samples increased with the decreasing size of the pore-forming agent, which is the opposite of the trend in porosity. With the increase in dimensions of the PS, the bending strength gradually decreased from 181.1 MPa to 150.4 MPa and sharply reduced to 101.4 MPa when PS100 was added. The amount and size of pores are the main factors that affect the bending strength of a ceramic membrane. The smaller the pore-forming agent, the more uniform the distribution of PS, which leads the close binding of alumina powder and high bending strength. A larger size of pore-forming agent (PS) easily leads to uneven distribution, which affects the distribution of formed pores and causes pore overlap. Large pores potentially lead to stress concentration in sintered ceramic and worsen its mechanical strength. 

Porous alumina ceramic was prepared by solid phase sintering, and the size of the pore-forming agent affected its microstructure and mechanical properties. It should be noted that the mechanical property of bending strength decreased with the increase in porosity. A balance between porosity and mechanical property was achieved when PS50 was added, where the density, porosity and bending strength were 3.199 g/cm^3^, 11.8%, and 150.4 MPa, respectively. 

### 3.3. Simulation Analysis and Experimental Verification

In Figure 2, the structure of the air bearing is set as follows: the thickness of the gas film (*h*) is 100 μm, porosity (*P*) is 11.8% and environmental temperature (*T*) is 300 K. Using ANSYS/Fluent software, the gas pressure on the bearing surface was analyzed while under different gas supply pressures. Figure 7 represents the gas pressure distribution on the bearing’s surface when the gas supply pressure (*P*s) is 0.1 MPa, 0.2 MPa, 0.3 MPa, 0.4 MPa and 0.5 MPa. It can be found from the pressure nephogram that the pressure on the bearing surface gradually decreases from the center to the surrounding. The gas pressure on the bearing surface increases with the gas supply pressure. The gas pressure on the bearing surface was 0.0147 MPa when the gas supply pressure was 0.1 MPa, while it achieved 0.0688 MPa at a gas supply pressure of 0.5 MPa. The gas pressure on the bearing surface increased with the gas supply pressure; that is, the bearing capacity of the proposed air bearing increased with the gas supply pressure, as shown in Figure 7f. At a gas film thickness of 100 μm and porosity of 11.8%, the bearing capacity of the air bearing increases from 8.6 N to 40.8 N when the gas supply pressure increases gradually from 0.1 MPa to 0.5 MPa.

In order to analyze the effect of gas supply pressure, an experiment was performed with an air compressor (Type.OTS-550, Taizhou Outstanding Industry and Trade Co. Ltd., Taizhou, China), a displacement sensor (Type. IL-S025, Keyence Corp., Osaka, Japan) and oscilloscopes (Type. TBS2000B, Tektronix Inc., OH, USA). The proposed air bearing was fabricated with an aluminum shell and sintered porous alumina ceramic membranes. The membrane was processed from the ceramic, and its parameters are listed in Table 2. After the membrane was bonded to the aluminum shell, the air bearing was connected to the air compressor via a pressure regulating valve and a plastic hose. The compressed air was sent to the air bearing through a pressure-regulating valve and flowed out of the porous alumina ceramic. With a certain load on the air bearing, its floating was measured by a displacement sensor, whose signal was acquired by oscilloscopes, as shown in Figure 8.

Experimental results indicate that the gas film thickness increases with the gas supply pressure at a fixed load and decreases with the increase of load at a fixed gas supply pressure. It should be noted from Figure 7 and Figure 9 that both the gas film thickness and gas supply pressure affect the bearing capacity of the proposed air bearing. When the gas supply pressure increases from 0.1 MPa to 0.5 MPa, the gas film thickness increases from 86μm to 215 μm under a load of 1 kg. The gas film thickness decreases with the increase of load. The maximum gas film thickness reaches 215 μm when the gas supply pressure and load are 0.5 MPa and 1 kg, respectively. The minimum gas film thickness reaches 18 μm when the gas supply pressure and load are 0.1 MPa and 4 kg, respectively. When the gas film thickness is 100 μm and the gas supply pressure ranges from 0.1 MPa to 0.5 MPa, the simulation results shown in Figure 7f match the experimental data in Figure 9. The accuracy of the conclusion is verified by the combination of simulation and experiment, which has reference and guiding significance for the design and optimization of air bearings.

## 4. Conclusions

A combination of optimization design and material modification was proposed to improve the performance of the air bearing. After air bearing modeling in SolidWorks software, flow field was analyzed by Fluent software. The results of the pressure nephogram indicated that the stable area of gas pressure was too small to support the load when the porosity of the porous ceramic membrane was less than 10%. With the addition of polystyrene microspheres as the pore-forming agent, the porous alumina ceramic membrane was sintered from alumina formula powder. In the porosity range between 6.2% and 25.4%, the mechanical property of sintered samples decreased with the porosity. When the ceramic membrane’s porosity was 11.8%, a balance between porosity and mechanical property was achieved. The proposed method was further verified using both simulation and experimental results, demonstrating the stability and effectiveness of the designed air bearing.

## Figures and Tables

**Figure 1 membranes-11-00872-f001:**
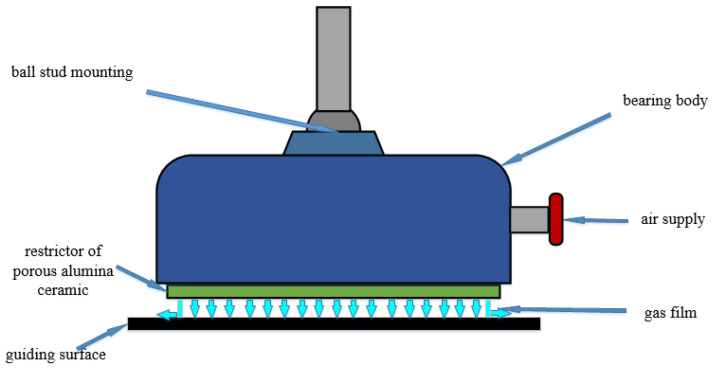
Principle of porous alumina ceramic membrane for the air bearing.

**Figure 2 membranes-11-00872-f002:**
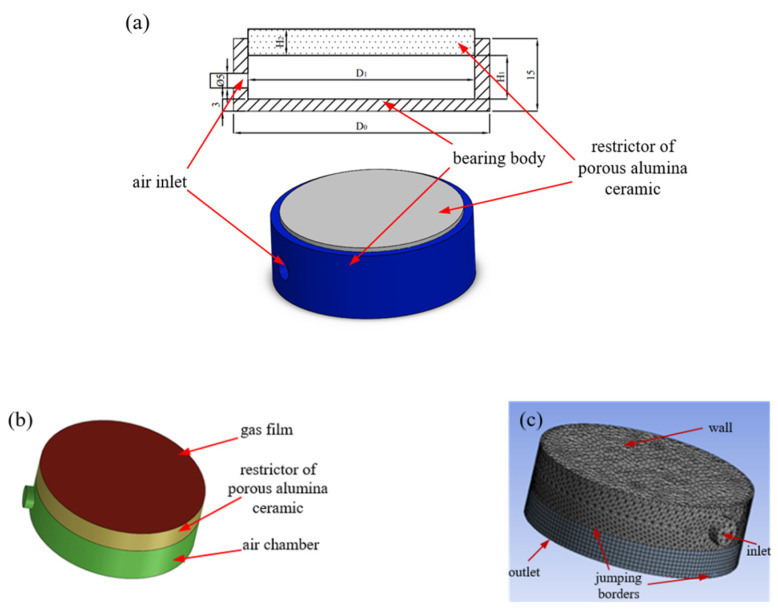
Simulation model of the proposed air bearing: (**a**) geometric model, (**b**) fluid model and (**c**) meshing.

**Figure 3 membranes-11-00872-f003:**
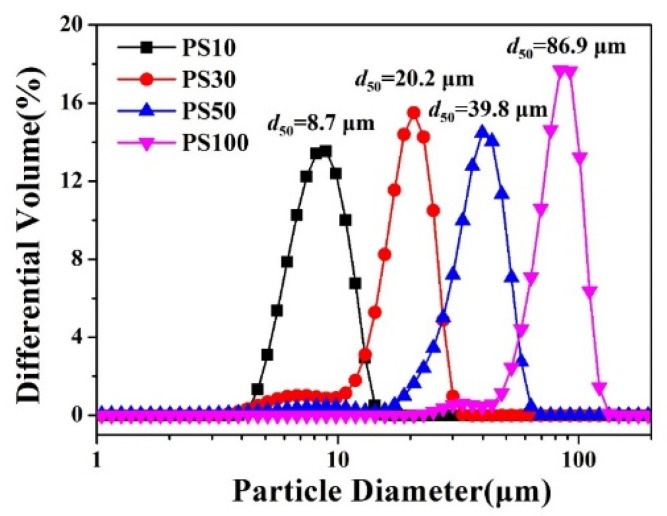
Particle distribution of the PS10, PS30, PS50 and PS100.

**Figure 4 membranes-11-00872-f004:**
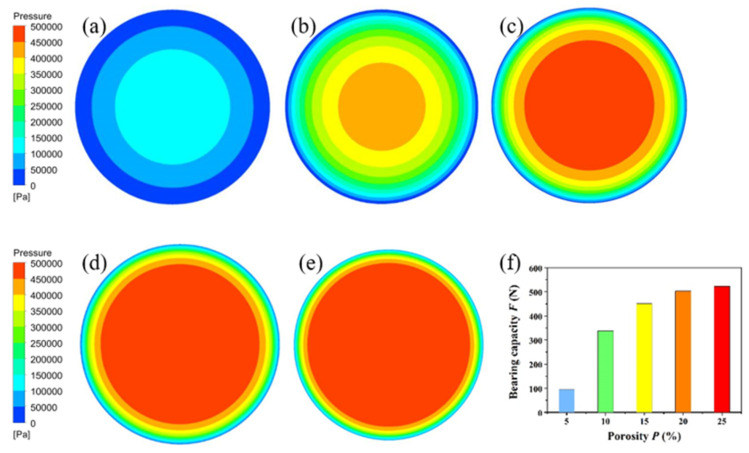
Gas pressure distribution on the bearing surface at different porosity: (**a**) 5%, (**b**) 10%, (**c**) 15%, (**d**) 20%, (**e**) 25% and (**f**) the relationship between bearing capacity and porosity.

**Figure 5 membranes-11-00872-f005:**
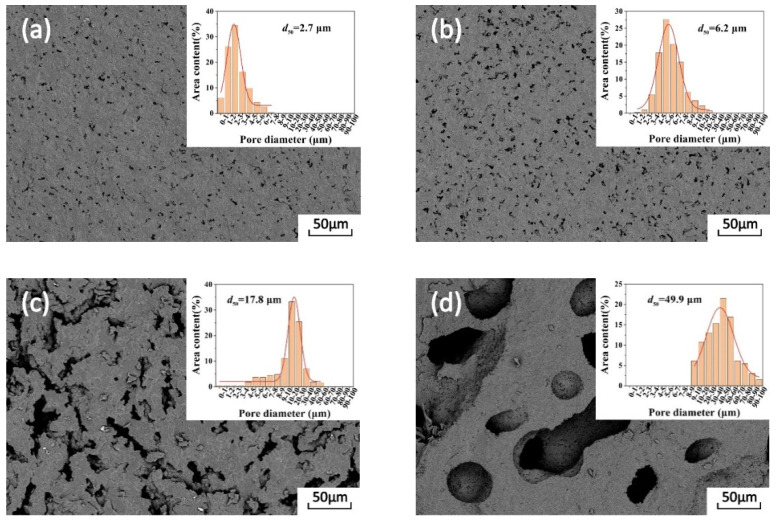
SEM images of porous Al_2_O_3_ ceramic membrane prepared by adding PS with different specifications: (**a**) PS10, (**b**) PS30, (**c**) PS50 and (**d**) PS100. The insets in (**a**–**d**) show the corresponding pore size distribution.

**Figure 6 membranes-11-00872-f006:**
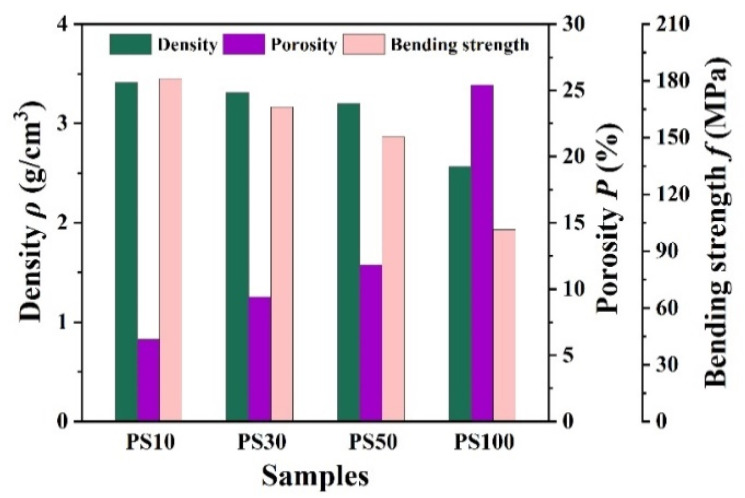
Density, porosity and bending strength of samples prepared by adding PS with different specifications.

**Figure 7 membranes-11-00872-f007:**
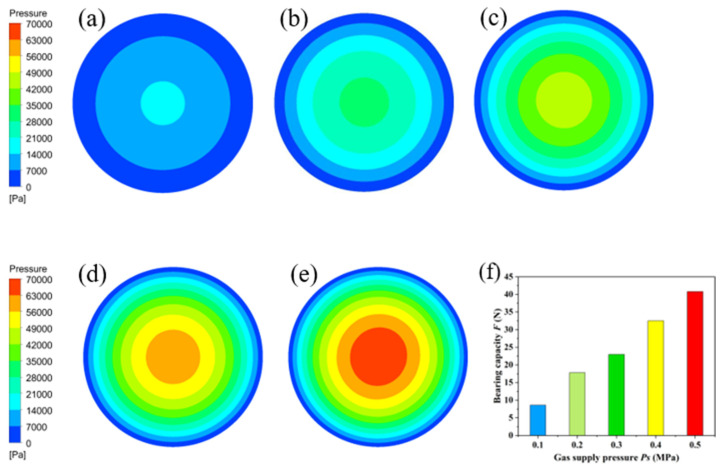
Distribution of gas pressure on the bearing’s surface with different air supply pressure: (**a**) 0.1 MPa, (**b**) 0.2 MPa, (**c**) 0.3 MPa, (**d**) 0.4 MPa, (**e**) 0.5 MPa and (**f**) the relationship between bearing capacity and gas supply pressure.

**Figure 8 membranes-11-00872-f008:**
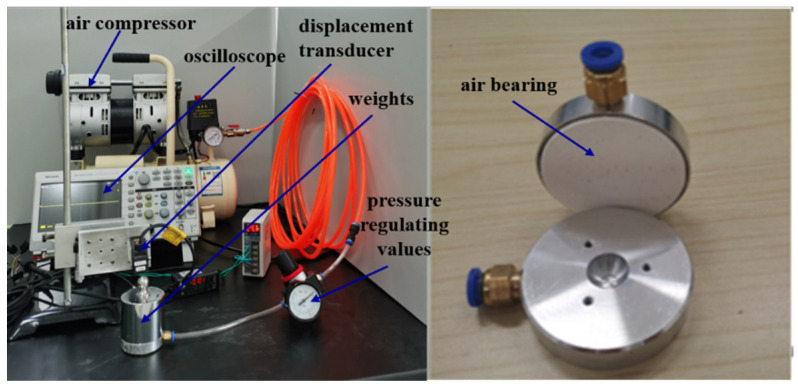
The static measurement system device of the air bearing.

**Figure 9 membranes-11-00872-f009:**
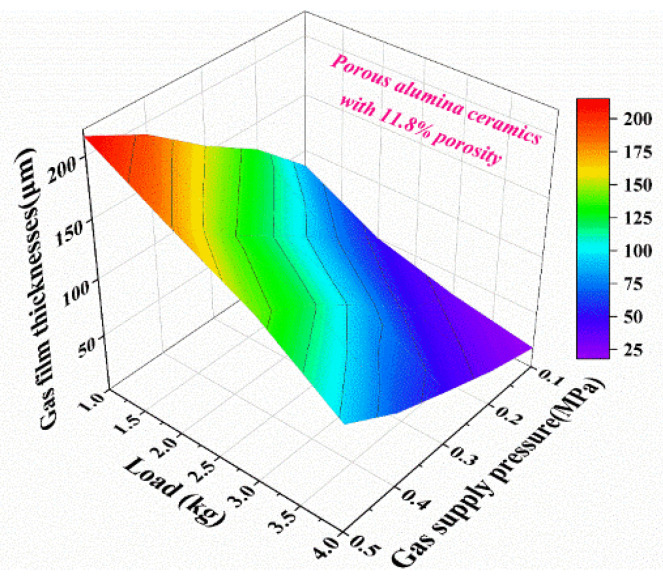
Gas film thicknesses at different gas supply pressure and load.

**Table 1 membranes-11-00872-t001:** Parameters of the air bearing.

Parameter	Value
Bearing body diameter, *D*_0_ (mm)	44
Gas space depth, *H*_1_ (mm)	8
Porous medium diameter, *D*_1_ (mm)	40
Porous medium thickness, *H*_2_ (mm)	5
Film thickness, *h* (μm)	25
Environment temperature, *T* (K)	300
Gas supply pressure, *P*s (MPa)	0.5
Environment stress, *P*o (MPa)	0.1
Dynamic viscosity of air, *η* (Ns/m^2^)	1.7894 × 10^−5^
Density of air, *ρ* (kg/m^3^)	1.225

**Table 2 membranes-11-00872-t002:** The static measurement of experimental parameters of the air bearing.

Parameter	Value
Porous medium diameter, *D*_1_ (mm)	40
Porous medium thickness, *H*_2_ (mm)	5
Porosity, *P* (%)	11.8
Environment stress, *P*o (MPa)	0.1
Gas supply pressure, *P*s (MPa)	0.1~0.5
Load, *M* (kg)	1~4

## Data Availability

Not applicable.

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
