# Peer review of "Rational Design and Porosity of Porous Alumina Ceramic Membrane for Air Bearing"

_membranes, 2021, doi:10.3390/membranes11110872_

Round 1
Reviewer 1 Report
Manuscript ID.: membranes-1408150
- The abstract should be revised to be more specific. Please add some important numerical results and describe them in more detail. Ex.) Line 19: much improved. How much?? And the abstract is too short to attract the interest of readers.
- The first paragraph of the introduction: It is too long and not well organized. Please rewrite it.
- Line 53: “In addition” is not necessary.
- Line 117: How did you measure the median particle size??
- For the instrument, the authors should provide the name of the production company and the location of the company. And Nano Measure software needs more specifications. Line 227. Please provide the specific information of the instruments provided in Line 227.
- Line 227: They should be provided in Materials and Methods.
- The experimental procedures for pore size and bending stress should be provided in materials and methods.
- The accuracy of the model compared to experimental data should be provided.
- The sensitivity analysis is necessary for model study.
Reviewer 2 Report
Review of the article entitled Rational Design and Porosity of Porous Alumina Ceramics Membrane for Air Bearing
Authors: Jianzhou Du, Duomei Ai, Xin Xiao, Jiming Song, Yunping Li, Yuansheng Chen, Luming Wang, Kongjun Zhu
The reviewed manuscript revealed the strategy of optimization design and material modification of air bearing.
In my opinion, the reviewed manuscript complements the current research conducted in this subject and it is consistent with the topic of Membranes journal. The Authors in the Introduction section clearly stated the novelty and the objective of their work. The paper is well written and organized in a concise form, and the Authors supported their hypothesis by proper explanations. I recommend the manuscript to be published in Membranes journal in present form.
Round 2
Reviewer 1 Report
The authors revised the manuscript by reflecting the reviewer's comments, and the quality of the manuscript was improved. However, I recommend you to check the minor one.
Please add the space between numbers and units (except %).
